# Medium Term Outcomes of TVT-Abbrevo for the Treatment of Stress Urinary Incontinence: Efficacy and Safety at 5-Year Follow-Up

**DOI:** 10.3390/medicina58101412

**Published:** 2022-10-08

**Authors:** Andrea Braga, Fabiana Castronovo, Anna Ottone, Marco Torella, Stefano Salvatore, Alessandro Ferdinando Ruffolo, Matteo Frigerio, Chiara Scancarello, Andrea De Rosa, Fabio Ghezzi, Andrea Papadia, Giorgio Caccia, Maurizio Serati

**Affiliations:** 1Department of Obstetrics and Gynecology, EOC, Beata Vergine Hospital, 6850 Mendrisio, Switzerland; 2Faculty of Biomedical Sciences, Università della Svizzera Italiana, 6900 Lugano, Switzerland; 3Department of Gyanecology, Obstetric and Reproductive Science, Second University of Naples, 80138 Naples, Italy; 4Department of Obstetrics and Gynecology, IRCSS San Raffaele Scientific Institute, 20132 Milan, Italy; 5Department of Obstetrics and Gynecology, ASST Monza, San Gerardo Hospital, 20900 Monza, Italy; 6Department of Obstetrics and Gynecology, Del Ponte Hospital, University of Insubria, 21100 Varese, Italy; 7Department of Obstetrics and Gynecology, EOC, Civico Hospital, 6900 Lugano, Switzerland

**Keywords:** trans-obturator tension-free vaginal tape, TVT-abbrevo, TVT-O, mid-urethral sling, stress urinary incontinence, female urinary incontinence

## Abstract

*Background and objectives*: Stress urinary incontinence (SUI) is the most common type of urinary incontinence, affecting approximately 46% of adult women. After failure of conservative treatment, the mid-urethral sling (MUS) is considered the most effective and safe surgical procedure for SUI. In 2012, Waltregny et al. introduced a new trans-obturator tension-free vaginal tape (TVT) procedure, named TVT-abbrevo (TVT-A). The aim of the present study is to evaluate the efficacy and safety of the TVT-A procedure in women with pure SUI at 5-year follow-up. *Materials and Methods*: All women who complained of pure SUI symptoms with concomitant urodynamic stress incontinence (USI) were prospectively enrolled and treated with the TVT-A procedure. Postoperative subjective outcome measures included: International Consultation on Incontinence Questionnaire–Short Form (ICI-Q SF), Patient Global Impression of Improvement (PGI-I) scale, and patient degree of satisfaction scale. A PGI-I score ≤ 2 and a patient-satisfaction score ≥8 were used to define subjective success. Objective success was defined as the absence of urine leakage during a cough stress test. Adverse events were collected according to the Clavien–Dindo classification during follow-up. *Results*: Univariable analysis was used to investigate outcomes. Fifty women who met the inclusion criteria underwent TVT-A implantation. At 5 years after TVT-A implantation, 38 out of 45 (84.4%) patients were subjectively cured (*p* for trend 0.05), and 40 out of 45 (88.9%) patients were objectively cured (*p* for trend 0.04). A significant trend of de novo OAB symptoms was reported (22.2% [10/45]) at the 5-year follow-up. No serious early or late complications such as urethral/bladder injury, persistent groin-thigh pain, and sexual dysfunction that required mesh removal were detected. The univariate analysis did not reveal any risk factors (i.e., age, body mass index (BMI), menopause, obstetric factors, and preoperative ICIQ- SF questionnaire) statistically associated with failure of the TVT-A procedure. *Conclusions:* In conclusion, the 5-year follow-up results of this study demonstrated that TVT-A is a safe and effective option for treatment of SUI with a very low rate of post-operative groin–thigh pain

## 1. Introduction

Stress urinary incontinence (SUI) is the most common type of urinary incontinence, affecting approximately 46% of adult women, peaking at 50% in women 40 years and older [1].

After failure of conservative treatment, the mid-urethral sling (MUS) is considered the most effective and safe surgical procedure for SUI [2,3]. However, its efficacy has recently been questioned due to reports of vaginal mesh complications. These criticisms failed to make a distinction between mesh for prolapse and mesh for incontinence. The guidelines of leading urological and urogynecological associations support the use of synthetic mesh MUSs in the treatment of SUIs with a strong level of evidence [4]. Nevertheless, more information on the long-term results and adverse effects of the mesh were needed [5], especially as a new surgical procedure was introduced.

In 2012, Waltregny et al. [6] introduced a modified trans-obturator tension-free vaginal tape (TVT) procedure, named TVT-abbrevo (TVT-A), with the aim of reducing the incidence of groin and thigh pain [7]. This technique avoids the perforation of the obturator membrane with a scissor and guide, reducing the depth of lateral dissection. Furthermore, the polypropylene tape is shorter than the traditional sling (12 cm long).

Several studies have shown that TVT-A reduces the rate of post-surgical pain and complications compared with the traditional transobturator approach [6,8].

Nevertheless, a small number of studies in the literature analyzed the efficacy and safety of TVT-A in the medium- and long-term. We previously reported the 3-year subjective and objective results of women who underwent TVT-A implantation [9] and demonstrated that it was a highly effective and safe option for the treatment of female SUIs.

This is an update of our previous study [9]. We evaluated women who underwent TVT-A procedure at a follow-up of at least 5 years.

## 2. Materials and Methods

Between June 2014 and December 2017, all women who were referred to our Urogynecological Department complaining of pure SUI symptoms were prospectively enrolled in our study. We only considered patients eligible for surgical treatment (Gynecare TVT Abbrevo^®^ Continence System; Ethicon Inc., Somerville, NJ, USA) when they had confirmed urodynamic stress incontinence (USI).

As reported in our previous study [9], our exclusion criteria included: previous radical pelvic surgery, neurological diseases, overactive bladder (OAB) symptoms, detrusor overactivity (DO), postvoid residual (PVR) > 100 mL, and concomitant pelvic organ prolapse (POP) greater than stage 1 according to the POP quantification (POP-Q) system [10].

The initial assessment included taking a medical history, physical evaluation, urinalysis, 3-day bladder diary, Q-tip test, and urodynamic study. Physical evaluations were performed with the patient in the lithotomy position and POP was detected during a maximal Valsalva maneuver, according to the POP-Q system [10]. The urodynamic study (UDS), including uroflowmetry, filling cystometry, urethral pressure measurement, and pressure/flow study, was performed in accordance with criteria established by the urodynamic practice guidelines of the International Continence Society [11]. We defined urethral hypermobility as a Q-tip angle of >30°. All definitions, methods, and units were based on the last version of the International Continence Society’s standardization of terminology [12]. In addition, all women completed the ICIQ-SF questionnaire [13]. For the surgical procedures, expert surgeons followed the original technique introduced by Waltregny et al. [6].

As previously published [14], spinal or general anesthesia was performed according to clinical features and/or patients preference [14]. Annual postoperative evaluations were scheduled until the fifth year of follow-up. Further evaluations were planned at the physician’s discretion. The following postoperative investigations were included: medical history, physical examination, 3-day bladder diary, Q-tip test, and evaluation of subjective satisfaction using validated questionnaires. An objective cure was defined as the absence of urine leakage during a stress test in the lithotomy and upright positions with a full bladder (>400 mL by ultrasound measurement).

To detect subjective outcomes, all women completed the ICIQ-SF, PGI-I scale [15], and a 10-point patient satisfaction scale (0 means ‘‘not improved’’ and 10 ‘‘the highest rate of improvement’’) [16]. PGI-I scores ≤2 and a patient-satisfaction score ≥8 were used to define subjective success, as previously reported by Abdel-Fattah et al. [17]. The Clavien–Dindo classification was used to report complications.

All women who complained of the onset of de novo OAB symptoms underwent a UDS. [18]. Antimuscarinic therapy (fesoterodine 4 mg or solifenacin 5 mg) was administered to patients with OAB for at least 12 weeks. The efficacy of this therapy was assessed using the previously described 3-point symptom assessment scale [19]. All women signed an informed consent form, and the Declaration of Helsinki was followed. The study did not need ethical/institutional review board approval because normal clinical practices were followed [20].

### Statistical Analysis

IBM-SPSS v.17 for Windows (IBM Corp, Armonk, NY, USA) was used to perform statistical analyzes. Continuous variables were reported in median and interquartile range. We used the χ^2^ test and χ^2^ test for trend to analyze and compare surgical outcomes during the five years of follow-up. The null hypothesis was that there would be no association between the success rate of TVT-A and time. A one-way analysis of variance was used to compare a continuous series of variables with the subjective outcome measures. A univariate analysis was performed using the Cox proportional hazards model to evaluate factors potentially affecting the risk of recurrence during the follow-up period. A *p*-value < 0.05 was considered to be statistically significant.

## 3. Results

A total of 50 women with urodynamic SUIs who met the inclusion criteria and underwent TVT-A implantation were enrolled in the study. Patient characteristics at baseline are reported in Table 1. Five patients were lost to follow-up for the final 5-year postoperative evaluation. Subjective and objective cure rates are summarized in Table 2. At 5 years after the TVT-A procedure, 38 out of 45 (84.4%) patients were considered subjectively cured (*p* for trend 0.05), and 40 out of 45 (88.9%) patients were considered objectively cured (*p* for trend 0.04). Among the failures, only 1 patient (2.2%) decided to undergo a second surgical procedure with a bulking agent (Macroplastique^®^). Onset of de novo OAB symptoms was reported by 4 (2/50), 14.2 (7/49), and 22.2% (10/45) at the 1-, 3- and 5-year evaluation, respectively. We recorded a statically significant increase in de novo OAB at the 5-year follow-up (*p* for trend 0.02). All these women presented with dry OAB and started antimuscarinic treatment. Twelve weeks later, 40% (4 of 10) were nonresponders. The trend of the subjective outcomes (ICI-Q—SF and PGI-I scores) over time is reported in Table 3. Long-term data showed no significant deterioration of subjective outcomes. Data regarding groin–thigh pain are reported in Table 4. Only 3 patients (6%) at 1 day after surgery reported pain with VAS ≥ 5 and the overall median VAS score was very low 1 (0–7). No patient mentioned pain either at the 1-month or 5-year follow-up. Long-term complications classified according to the Clavien–Dindo classification are summarized in Table 5. As reported at the 3-year follow-up [9], only 2 patients had vaginal mesh exposure (1 asymptomatic) that was treated with topical estrogen followed by surgical intervention (vaginal mucosa dissection and mesh covering using vaginal tissue) without mesh excision. One woman complained of the persistence of voiding dysfunction, which did not require tape release or resection during the follow-up period. No other serious complications (urethral/bladder injury or erosion and dyspareunia) were recorded. No risk factors (age, BMI, menopause, obstetric factors, and preoperative ICIQ- SF questionnaire) were statistically associated with the recurrence of SUI in the univariate analysis.

## 4. Discussion

This study showed that TVT-A is an appropriately effective and safe option for the treatment of SUIs at medium-term follow-up. We found 5-year subjective and objective cure rates of 84.4 and 88.9%, respectively. Furthermore, low complications were registered and only one patient needed a second surgical procedure.

Despite the notice issued by the United States Food and Drug Administration (FDA) considering MUSs to be relatively safe [21], their role in the treatment of female SUIs has been recently debated, due to reports of vaginal mesh complications.

TVT-A is one of the newest MUS devices for the treatment of SUIs, introduced by Waltregny et al. [6]. Several studies have produced extremely encouraging results at short-term evaluation [22,23,24]; however, only a few studies in the literature have assessed the medium-long term durability and safety of this surgical procedure.

Capobianco et al. [25] found that, in a cohort of 56 patients who underwent TVT-A for moderate-severe SUIs, 43 out of 56 patients (76.79%) had a postoperative urodynamic (12-month follow-up) that was normal, and it was significantly improved in 10 of the 56 patients (17.86%). At two years follow-up, 94.64% of patients were considered subjectively cured or markedly improved.

Another prospective study on 76 patients, by Kurien et al. [26], analyzed the outcomes of the TVT-A procedure over a period of 22 months. The subjective cure rate for SUI at one-year follow-up was 90.6%. Among the nine patients who completed the two-year follow-up, 88.9% reported either cure or improvement of SUI. The objective cure rate was 86.8% at six months, with a low rate of complications and groin pain. Zullo et al. [27], in a prospective study, compared the efficacy, safety, and complications of the TVT-O and TVT-A in the treatment of female SUIs. Overall, 138 of 158 patients (87%) were objectively cured from SUIs 36 months after the operation, with no significant differences found between groups [69 (87%) and 69 (87%) patients for the TVT-O and TVT A, respectively]. Urodynamic studies that were performed 3 years postoperatively demonstrated an equal objective cure rate in both groups, although they were lower compared with the subjective cure rate (preoperative value of ICIQSF-UI in TVT-O group was 7.8 + 4.8 vs. 3-year value 2.4 + 1.6, *p* < 0.01, whereas the preoperative value of ICIQSF-UI in the TVT-A group was 7.7 + 3.9 vs. 3-year value 2.7 +1.8, *p* < 0.01).

Recently, in a retrospective study, Ruffolo et al. [28] compared objective and subjective outcomes in women with SUIs, who either had a TVT-A or single incision mini-sling (SIMS) ALTIS, at 5-year follow up. Forty-two patients were evaluated in the TVT-A group and 58 in the ALTIS group. No significant differences were found in subjective (88.1 vs. 89.7%, *p* = 0.806) and objective (81.0 vs. 86.2%, *p* = 0.479) cure rates between the two groups. The incidence of long-term complications was similar in both groups, mainly classified as Dindo II grade (de novo OAB) and conservatively managed.

Exposure of the sling (Dindo IIIa grade) occurred in 2.4% (1/42) of TVT-A at 30 months from implantation, whereas it occurred in 5.2% (3/58) of SIMS-ALTIS procedures at 31, 33, and 35 months from surgery (*p* = 0.482). Three (3/4; 75%) of them were covered, whereas one was partially removed (1/4; 25%).

For the first time in the literature, this study reports a prospective subjective and objective 5-year evaluation of the TVT-A procedure.

Our findings suggest that TVT-A is a safe and highly effective procedure for the treatment of SUIs, even in mid-term and long-term follow-up. Although not statistically significant, there seemed to be a slight trend of efficacy reduction with time (from over 95 to 84–88%), and the long-lasting benefits seem to be slightly lower than traditional MUSs (TVT, TVT-O) [2,3,29]; however, more studies with a large sample size are needed to demonstrate this.

As previously reported at the 3-year follow-up [9], the trend of de novo OAB was significantly increased at the final 5-year evaluation. The rate of the onset of de novo OAB did not differ from the traditional MUSs [2,3,29], and it is comparable with previous published data on the TVT-A [27]. Although we know that antimuscarinic treatment of patients with de novo OAB after MUS implantation is associated with lower efficacy [19], in our series, patients who started an antimuscarinic treatment for de novo OAB seemed to respond slightly better than with TVT-O [29]. These data, which were sufficient for adequate preoperative counselling, needed to be confirmed. Also, patients reported significantly less post-operative pain than with traditional TVT-O. In fact, groin–thigh pain was quite common during the first 24–28 h (62–24%); this resolved in the short term as there were no patients with pain at 1 month. Most likely, the shorter tape reduced the injury/inflammation of the adductor muscles compared with the traditional sling, avoiding persistent pain. In our series, mesh-related complications appeared during the third year of follow-up [9], without need for excision.

As previously reported [9], the strengths of the present study include: (1) homogeneity of the population; (2) exclusion of patients with mixed incontinence and/or DO; (3) exclusion of patients who underwent previous anti-incontinence surgery and/or other concomitant surgical procedure; (4) use of validated instruments to detect subjective and objective outcomes; and (5) low number of patients lost to follow-up. The limitations of this study are its small sample size and lack of evaluation of the pad test

## 5. Conclusions

The 5-year results of this study demonstrated that TVT-A is a highly effective option for the treatment of SUI, with a very low rate of complications and groin–thigh pain, even if this procedure is associated with a slight but significant decrease in efficacy over time. However, the lack of long-term data prevents TVT-A from being considered as effective as the traditional trans-obturator sling. Further studies are needed to support its long-term efficacy and safety.

## Figures and Tables

**Table 1 medicina-58-01412-t001:** Patient characteristics at baseline.

	** *n* ** **= 50**
Age, year, median, (IQR)	54.6 (48–62)
BMI, kg/m2, median, (IQR)	26 (23–28)
Obese, BMI ≥ 30, no. (%)	7 (14)
Menopausal, no. (%)	28 (56)
HRT, no. (%)	7 (14)
Previous vaginal deliveries, median, (IQR)	2 (1–2.25)
Macrosome, ≥4000 g, no. (%)	6 (12)
Operative delivery, vacuum/forceps, no. (%)	8 (16)
Cesarean delivery, no. (%)	10 (20)
Previous hysterectomy, no. (%)	12 (24)
Smoking habits, no. (%)	15 (30)

BMI: body mass index; HRT: hormonal replacement therapy; IQR: interquartile range.

**Table 2 medicina-58-01412-t002:** Analysis of cure rates across the study period.

	1 Year	2 Year	3 Year	5 Year	*p* Value
Subjectively cured, no. (%)	48/50 (96)	46/50 (92)	44/49 (89.8)	38/45 (84.4)	0.27 a 0.05 b
Objectively cured, no. (%)	49/50 (98)	48/50 (96)	45/49 (91.8)	40/45 (88.9)	0.25 a 0.04 b
De novo OAB, no. (%)	2/50 (4)	3/50 (6)	7/49 (14.2)	10/45 (22.2)	0.02 a 0.002 b

OAB: overactive bladder; a: chi-squared test; b: chi-squared test for trend.

**Table 3 medicina-58-01412-t003:** Subjective outcome scores over time after TVT-A procedure.

		1 Year	2 Year	3 Year	5 Year	*p* Value
“Very much better” or “much better” on PGI-I scale, n/N (%)		48/50 (96)	46/50 (92)	44/49 (89.8)	38/45 (84.4)	0.27 a 0.05 b
Median (IQR) Patient satisfaction questionnaire score		10 (8–10)	10 (8–10)	10 (8–10)	10 (7–10)	0.88 *
Median (IQR) ICIQ-SF score	18 (14–18)	0 (0–6)	0 (0–6)	0 (0–6)	0 (0–8)	0.001 *

ICIQ-SF: International Consultation on Incontinence Questionnaire-Short Form; IQR: interquartile range; PGI-I: Patient Global Impression of Improvement; a: chi-squared test; b: chi-squared test for trend. *One-way ANOVA.

**Table 4 medicina-58-01412-t004:** Groin–thigh pain after TVT-A procedure.

	Day 0	Day 1	1 Month	5 Year	*p* Value
Patients with pain, no. (%)	31/50 (62)	12/50 (24)	0/49 (0)	0/45 (0)	<0.0001 a <0.0001 b
Pain, VAS score, median (IQR)	2 (0–8)	1 (0–7)	0 (0)	0 (0)	0.04 *
Patients with pain VAS ≥ 5, no. (%)	4/50 (8)	3/50 (6)	0/49 (0)	0/45 (0)	0.06 a 0.01 b

VAS: Visual Analogue Scale; a: chi-squared test; b: chi-squared test for trend. * One-way ANOVA.

**Table 5 medicina-58-01412-t005:** Clavien–Dindo classification of long-term complications.

Complication	*n* (%)	Action
Clavien grade I,		
Persitence of voiding dysfunction	1 (2)	Observation
Clavien grade II,		
Recurrent UTIs.	2 (4)	Antimicrobial prophylaxis or therapy
De novo OAB	10 (22.2)	Antimuscarinics/ß-agonists
Clavien grade III,		
Asymptomatic vaginal mesh exposure	1 (2)	Topical estrogen plus surgical intervention without mesh excision
Symptomatic vaginal mesh exposure	1 (2)	Topical estrogen plus surgical intervention without mesh excision

## Data Availability

All the data are available from the corresponding author upon reasonable request.

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
