# Peer review of "Medium Term Outcomes of TVT-Abbrevo for the Treatment of Stress Urinary Incontinence: Efficacy and Safety at 5-Year Follow-Up"

_medicina, 2022, doi:10.3390/medicina58101412_

Round 1

Reviewer 1 Report

Dear authors, i congratulate with your work. The paper is interesting, dealing with medium (not long) term results of TVT-A sling for female stress urinary incontinence. The article is well written and easy to be read. In my opinion, there are some issues that would require clarification:

- the abstract lacks the definition of "cured" patients, the definition of "serious" (a Clavien Dindo grade?) and the list of risk factors analyzed

- the introduction lacks a short description of the main differences between the two surgical techniques, it may be interesting for the reader to better understand the amelioration of TVT-A technique 

- did you standardize urodynamic SUI evaluation (filling volume, trigger maneuver - cough or Valsalva)? Many patients do not leak during urodynamics but have a clinically proven SUI (for example they leak when jumping, walking, doing physical efforts...): may the selection of only urodynamic-proven SUI represent a selection bias?

- does an in-office negative stress test always correlate to the condition of being completely dry? does the stress test always mimic the condition that makes the woman leak? do you have data about the pad test?  

- as indicated in the limitations, a 45 patients cohort enrolled in almost 4 years is a small population

- although not statistically significant, there seems to be a slight trend of efficacy reduction with time (from more than 95% to 84-88%); I would include this in the discussion section

- table 1: BMI line lacks the IQR interval 

Author Response

Reviewer 1

Dear authors, I congratulate with your work. The paper is interesting, dealing with medium (not long) term results of TVT-A sling for female stress urinary incontinence. The article is well written and easy to be read. In my opinion, there are some issues that would require clarification:

- the abstract lacks the definition of "cured" patients, the definition of "serious" (a Clavien Dindo grade?) and the list of risk factors analysed

We added these sentences in the  text:

“The PGI-I ≤ 2 and a patient-satisfaction score ≥ 8 were used to define subjective success. Objective succes was defined as the absence of urine leakage during the cough stress test.”

“No serious early and late complications such as urethral/bladder injury, persistent groin-thigh pain and sexual dysfunction that required mesh removal, were detected.”

“Risk factors analysed were: age, BMI, menopausal, obstetric factors and preoperative

 ICI-Q SF questionnaire”.

- the introduction lacks a short description of the main differences between the two surgical techniques, it may be interesting for the reader to better understand the amelioration of TVT-A technique

We added as follows in the text:

This technique, avoids the perforation of the obturator membrane with scissor and guide, reducing the depth of lateral dissection. Furthermore the polypropylene tape is shorter than traditional sling (12 cm long).

- did you standardize urodynamic SUI evaluation (filling volume, trigger maneuver - cough or Valsalva)? Many patients do not leak during urodynamics but have a clinically proven SUI (for example they leak when jumping, walking, doing physical efforts...): may the selection of only urodynamic-proven SUI represent a selection bias?

We used a standardized protocol according with criteria established by the good urodynamic practice guidelines of the International Continence Society. Additionally, we used to perform 1-3-5 cough stress test [Grigoriadis et al. The "1-3-5 cough test": comparing the severity of urodynamic stress incontinence with severity measures of subjective perception of stress urinary incontinence Int Urogynecol J 2016 Mar;27(3):419-25], and other clinical maneuvers (jumping, walking, etc…) to evaluate SUI. All patients included in our study leaked urine during UDS (without further maneuvers), so it is not represent a selection bias.

- does an in-office negative stress test always correlate to the condition of being completely dry? does the stress test always mimic the condition that makes the woman leak? do you have data about the pad test?

We know that stress test is not always correlate to the condition of being completely dry but unfortunately we don’t have data about the Pad-test (which is not a validate tool to detect SUI condition). However our subjective outcomes evaluation was made by validate questionnaires, which could reflect the real daily condition of SUI patients.

- as indicated in the limitations, a 45 patients cohort enrolled in almost 4 years is a small population

We know that the small cohort is a limitation, however this study reports, for the first time in the literature, a prospective subjective and objective 5-year office evaluation after TVT-A procedure

- although not statistically significant, there seems to be a slight trend of efficacy reduction with time (from more than 95% to 84-88%); I would include this in the discussion section

We added this sentence in the discussion

- table 1: BMI line lacks the IQR interval 

We added IQR in the table 1

Reviewer 2 Report

Thank you for asking me to review the study entitled “Medium term outcomes of TVT-Abbrevo for the treatment of Stress Urinary Incontinence: efficacy and safety at 5 years follow-up.”

The study is very interesting and adds valuable evidence for both safety and efficacy of the TVT-A approach. 

It is well written with an appropriate patient selection, exclusion criteria, number of subjects included, presented results and an interesting Discussion.

A couple of points:

Results

Line 145 could you please clarify what was the surgical intervention? You mention that it treatment was with topical estrogen without mesh excision. 

Discussion

I think it would be interesting to discuss briefly the fact that although groin-thigh pain was quite common during the first 24-28 hours (62-24%) this resolved in the short term as there were no patients with pain at 1 month. This is a very useful point not only during patient counselling but it also reinforces the evidence that transobturator slings are safe. Could the authors make any hypothesis of why the design of TVT-A has such low % of postoperative groin pain? 

Author Response

Reviewer 2

The study is very interesting and adds valuable evidence for both safety and efficacy of the TVT-A approach. 

It is well written with an appropriate patient selection, exclusion criteria, number of subjects included, presented results and an interesting Discussion.

A couple of points:

Results

Line 145 could you please clarify what was the surgical intervention? You mention that it treatment was with topical estrogen without mesh excision. 

The surgical intervention was: dissection of the vaginal mucosa and covering of the mesh  without its removal, using the dissected vaginal tissue. We added this last sentence in the text.

Discussion

I think it would be interesting to discuss briefly the fact that although groin-thigh pain was quite common during the first 24-28 hours (62-24%) this resolved in the short term as there were no patients with pain at 1 month. This is a very useful point not only during patient counselling but it also reinforces the evidence that transobturator slings are safe. Could the authors make any hypothesis of why the design of TVT-A has such low % of postoperative groin pain? 

We added these sentences in the text:

In fact, groin-thigh pain was quite common during the first 24-28 hours (62-24%) this resolved in the short term as there were no patients with pain at 1 month.

Probably, the shorter tape than the traditional sling could reduce the injury/inflammation to the adductor muscles avoiding persistent pain.

Round 2

Reviewer 1 Report

Dear Authors, thank you for your reply. I appreciate that you have taken my comments into consideration. The article is written fluently, the statistical analysis is correct and the results are well present.

I would highlight a persisting study design issue. A subjective evaluation with questionnaires on a limited (n=45) population can hardly represent the overall objective result of a surgical technique. The choice of the questionnaire for a subjective evaluation is correct. However, when it comes to incontinence, the subjective component depends on many variables (education, severity of incontinence, age, social status ...) and this variability may expose to potentially wrong interpretations of the results. A stress test was chosen as an objective parameter, but it does not always determine a urinary leakage in patients and it cannot be used as a universal criterion. The lack of evaluation of the pad test (a tool recommended by several scientific societies, expert opinion and cited by the guidelines and by most of the papers on SUI) in clinical practice represents an important limitation of this study.

Author Response

Dear Reviewer,

Thank you for your supportive comment.

We agree with you regarding the lack of the pad test evaluation.

We have pointed out this limitation in the discussion